A novel similarity score based on gene ranks to reveal genetic relationships among diseases

Luo Dongmei 1 2
Zhang Chengdong 3
Fu Liwan 1
Zhang Yuening 4
http://orcid.org/0000-0002-5730-0317 Hu Yue-Qing 1 5 yuehu@fudan.edu.cn
1 State Key Laboratory of Genetic Engineering, Institute of Biostatistics, School of Life Sciences, Fudan University , Shanghai , China
2 Department of Information and Computing Science, School of Mathematics and Physics, Anhui University of Technology , Ma’anshan, Anhui Province , China
3 Shanghai Public Health Clinical Center, Fudan University , Shanghai , China
4 SJTU-Yale Joint Center for Biostatistics, Shanghai Jiao Tong University , Shanghai , China
5 Shanghai Center for Mathematical Sciences, Fudan University , Shanghai , China
Bolshoy Alexander
Electronic publication date: 2021 Jan 6
Publication date: 2021
Volume: 9
Electronic Location ID: e10576
Received 2020 Jul 16; Accepted 2020 Nov 24
Copyright: © 2021 Luo et al.
Copyright year: 2021
Copyright holder: Luo et al.
License: This is an open access article distributed under the terms of the Creative Commons Attribution License, which permits unrestricted use, distribution, reproduction and adaptation in any medium and for any purpose provided that it is properly attributed. For attribution, the original author(s), title, publication source (PeerJ) and either DOI or URL of the article must be cited.
License URL: https://creativecommons.org/licenses/by/4.0/

Keywords: Gene expression, Similarity score, Genetic relationships, Disease network, Gene ranks

Funding: National Natural Science Foundation of China 11971117 and 11571082 Natural Science Foundation of Anhui Province of China 1808085MG220 This work was supported by the National Natural Science Foundation of China (grants nos. 11971117 and 11571082) and the Natural Science Foundation of Anhui Province of China (grant no. 1808085MG220). The funders had no role in study design, data collection and analysis, decision to publish, or preparation of the manuscript.

==============================
Knowledge of similarities among diseases can contribute to uncovering common genetic mechanisms. Based on ranked gene lists, a couple of similarity measures were proposed in the literature. Notice that they may suffer from the determination of cutoff or heavy computational load, we propose a novel similarity score SimSIP among diseases based on gene ranks. Simulation studies under various scenarios demonstrate that SimSIP has better performance than existing rank-based similarity measures. Application of SimSIP in gene expression data of 18 cancer types from The Cancer Genome Atlas shows that SimSIP is superior in clarifying the genetic relationships among diseases and demonstrates the tendency to cluster the histologically or anatomically related cancers together, which is analogous to the pan-cancer studies. Moreover, SimSIP with simpler form and faster computation is more robust for higher levels of noise than existing methods and provides a basis for future studies on genetic relationships among diseases. In addition, a measure MAG is developed to gauge the magnitude of association of anindividual gene with diseases. By using MAG the genes and biological processes significantly associated with colorectal cancer are detected.

Introduction

Exploring the common genetic basis of complex human diseases is often useful for understanding the disease relationships drawing on their genetic mechanisms. A similarity measure is a central component in detecting the common genetic basis among different diseases. Several common metrics have been proposed to measure similarities between diseases, such as Pearson, Spearman correlation coefficient, Euclidean distance, Manhattan distance, and Jaccard correlation coefficient (Antosh et al., 2013; Dennis et al., 2003; Serra, Romualdi & Fogolari, 2016; Shi, Yi & Ma, 2014). However, with the technological developments in molecular biology, large-scale gene expression profiling datasets produced from diverse technological platforms necessitate new and adaptive similarity measures to reveal meaningful genetic relationships across multiple platform types. Ranking genes according to their contribution to each disease can convert heterogeneous data into platform-independent rank lists, and a recent study (Serra, Romualdi & Fogolari, 2016) suggested that the discrimination ability of the similarity/distance measures based on ranked gene lists perform well or better than traditional measures (such as Euclidean distance and correlation coefficient), particularly for gene expression datasets produced with different biotechnologies (microarray, RNA-seq, etc.). Therefore, the genetic overlaps among diseases can be detected by integrating multi-platform datasets using similarity measures based on ranked gene lists, which can help us gain further insight on understanding disease etiology.

A review of related studies in similarity measures based on ranked gene lists showed that most of them depend on either a fixed cutoff position to consider overlaps between the top part of ranked gene lists or variational cutoffs to select the one that generates the most significant results (Chen, Zhou & Sun, 2015; Dennis et al., 2003; Von Mering et al., 2006). To avoid the uncertainty of results owing to arbitrariness in cutoff settings, global rank-based similarity measures have been developed. For example, the algorithm of GOrilla is a tool to discover and visualize the enrichment of GO terms that ranks genes by fold-change (Eden et al., 2009). The algorithm of gene set enrichment analysis (GSEA) allows all genes to contribute to overlapping signals in proportion to their degree of differential expression and can detect the weak signals that would be discarded by “threshold” approaches (Efron & Tibshirani, 2007; Subramanian et al., 2005). The algorithm of CORaL estimates the significant set size using the overlaps between sections of the ranked gene lists by maximizing statistical likelihood (Antosh et al., 2013). The algorithm of R2KS particularly emphasizes finding the same items near the top of the ranked gene list (Ni & Vingron, 2012). Plaisier et al. (2010) presented a popular “threshold-free” approach in neuroscience and biomedicine named rank-rank hypergeometric overlap (RRHO), which identifies and visualizes regions of significant overlap between two ranked gene lists and determines the statistical significance of enrichment by hypergeometric distribution.

Moreover, a similarity measure OrderedList that focuses on evaluating whether there is significant overlap between two ranked gene lists was proposed (Yang et al., 2006). The OrderedList is the weighted sum of the number of overlapping genes with an exponentially decaying weight, where a parameter β is introduced to determine how deep to go in the ranked gene lists. There are many studies on statistics-based improvement from similarity measures OrderedList. For example, Serra, Romualdi & Fogolari (2016) introduced similarity measure FESβ (fraction enrichment sum) and set the exponentially decaying parameter β as 0, 0.001, and 0.01. Chen, Zhou & Sun (2015) adopted the default series of β values in the R package OrderedList and proposed a robust statistic minimum p value: minβ⁡pβ, where parameter β can be calculated by setting a minimum weight and a default series of positions. To avoid arbitrariness and manual intervention of parameter β selection, Chen, Zhou & Sun (2015) finally found a parameter-free similarity measure WeiSumE* with good performance to detect overlaps among ranked gene lists in simulations, which is the weighted sum that normalizes the number of overlapping genes on the top genes of two ranked gene lists by its expectation. Generally, existing rank-based similarity measures are mostly based on the overlaps among ranked gene lists and may lead to information loss owing to fixed cutoff positions in the ranked gene lists, the uncertainty of results because of the arbitrariness of the cutoff position, or heavy computational burden.

In this study, we propose MAG based on the transformation of gene ranks to measure the magnitude of association of the individual gene with two diseases. By exploring the summation of MAG over all genes, we develop a novel similarity score SimSIP with simpler form and light computation burden to gauge genetic overlap among diseases based on gene ranks instead of intersections between the top part of ranked gene lists. To show the superiority of SimSIP, we firstly conduct a series of simulation studies to demonstrate the performance of SimSIP compared to some existing similarity measures based on ranked gene lists (WeiSumE* (Chen, Zhou & Sun, 2015), OrderedList (Yang et al., 2006), FES0.001, FES0.01(Serra, Romualdi & Fogolari, 2016)) and Euclidean distance EucD under various scenarios. Secondly, we apply SimSIP to analyze the gene expression data of cancers in The Cancer Genome Atlas database and find that it sheds light on the genetic relationships among cancers. Thirdly, we arrange the significantly similar cancer pairs among the 18 cancer types detected by SimSIP into a disease network in which the tendency to cluster the histologically or anatomically related cancers provides basic support for pan-cancer studies. Finally, for the most significantly similar cancer pair, colon adenocarcinoma (COAD) and rectum adenocarcinoma (READ), found by SimSIP, we use MAG to measure the magnitude of association of each gene both with COAD and READ and find the important oncogenes of colorectal cancer which are associated with COAD and READ and regulated in the same pattern. Moreover, biological processes highly associated with colorectal cancer are detected.

Materials and Methods

MAG and SimSIP

Let us assume that there are n genes for disease 1 and disease 2. For gene i, 1 ≤ i ≤ n, let ai be its rank among the n genes for disease 1, and bi is defined similarly for disease 2. Now we intend to gauge the genetic similarity or genetic relationship between these two diseases based on gene rank lists {ai}i=1n and {bi}i=1n. The gene rank usually represents the strength of association with the disease, in the sense of small rank meaning strong association and big rank meaning weak association. For example, the rank can be assigned for each gene by the magnitude of p value reported in the study of detecting differentially expressed genes, which is the routine work in the literature. To facilitate the construction of the similarity score, we transform the ranks ai and bi to their reciprocal 1/ai and 1/bi, whose values fall in the interval (0, 1] and can be treated directly as the magnitude of association of gene i with diseases 1 and 2, respectively. Further, we employ the geometric mean 1/ai⋅1/bi of 1/ai and 1/bi and call it MAG, which is a compromise between 1/ai and 1/bi as the magnitude of association of gene i with both diseases 1 and 2. Intuitively, a small value of MAG means a weak association of gene i with these two diseases, a big value means a strong association of gene i with these two diseases. Note the value of MAG is between n−1 and 1.

Now let us explore the summation ∑i=1n1/ai⋅1/bi of MAG over all genes, that is, the total magnitude of association for all n genes with diseases 1 and 2. Note that this summation is actually the inner product of vectors {1/ai}i=1n and {1/bi}i=1n with positive components. It is easy to check that the length of each of these two vectors is ∑i=1n1/i, which depends only on n and is independent of genes’ concrete ranks. So the summation is proportional to the Angle cosine between the two vectors mentioned above. Recall that the inner product in algebra or geometry theory is sometimes called the scalar product or dot product and is the projection of one vector on another in geometry space, which is a symmetric measure of closeness of two vectors. So the similarity score by inner product SimSIP=∑i=1n1/ai⋅1/bi

is an appropriate candidate for measuring similarity between contributions of all n genes to diseases 1 and 2, and it can be further taken as a similarity score between two diseases derived from the corresponding gene rank lists. Considering one extreme case in which ai = bi for every gene i, which means the rank of every gene for disease 1 is exactly equal to the rank of same gene for disease 2, these two diseases are the most similar in terms of these n genes and SimSIP attains its maximum ∑i=1n1/i. Considering the other extreme case in which the gene ranking top for disease 1 would always rank bottom for disease 2, and gene ranking bottom for disease 1 would always rank top for disease 2, these two diseases are the most dissimilar in term of these n genes and SimSIP attains its minimum. So a big value of SimSIP would imply that two diseases are similar.

Assessment of significance

It is easily observed from the expression of SimSIP that the more identically ranked genes associated with diseases 1 and 2 there are, the more similar two diseases are, and the larger SimSIP is. Therefore, an observed SimSIP larger than expected under the null hypothesis that two diseases have non-overlapping genes means significant. When we say that two diseases have no overlapping genes in the genetic perspective, we mean that the two gene rank lists are two random shufflings of 1∼ n. Based on the expression of SimSIP, we describe a procedure to generate the empirical distribution of SimSIP under the null hypothesis, which is used to evaluate the significance of the similarity score. Without loss of generality, we fix the first gene rank list as {i}i=1n and random permutate 1∼ n as {bi}i=1n for S times, then we obtain the corresponding SimSIPs, s = 1,2,…,S. The p value of SimSIP is computed as (1) p=∑s=1SI(SimSIPs≥SimSIP)S,

where I(·) is the indicator function assigning the value 1 or 0 relying on whether the condition within brackets is met. Let pr denote the p value and r = 1,2,…, R for R replications, and the power under the alternative hypothesis or the type I error under the null hypothesis for a given significance level α is (2) Power=∑r=1RI(pr≤α)R.

Regarding MAG, we can employ a similar procedure to evaluate the significance of an observed MAG. Under the null hypothesis that two diseases having non-overlapping genes, the two gene ranks ai and bi involved in are randomly drawn from {1,2,…, n} with replacement, and it is not difficult to get the distribution of MAG under the null hypothesis. The significance of the observed MAG can then be calculated.

Results

Simulation study

Parameter setting

We carry out simulations to evaluate the performance of SimSIP and the existing rank-based methods WeiSumE* (Chen, Zhou & Sun, 2015), OrderedList (Yang et al., 2006), FES0.001, and FES0.01 (Serra, Romualdi & Fogolari, 2016), which are weighted sum of the number of overlapping genes between the top part of ranked gene lists, and Euclidean distance EucD between the original values, which is a method without ranking.

As done in Chen, Zhou & Sun (2015), we set the number of genes n = 6,000, 11,000, and 25,000 and randomly choose two sets of d genes from the n genes as the associated genes with diseases 1 and 2, respectively. We fix the d = 1, 5, 10, and 20 in the simulation. The number o of overlapping associated genes in the two sets is taken as 0 for evaluating the type I error rate and positive for evaluating the power. For convenience, genes 1 to d are assumed to be associated with disease 1 and genes d − o + 1 to 2d − o are associated with disease 2, where d ≥ o. As performed by Chen, Zhou & Sun (2015), two normal distributions N(0,σ2) and N(1,σ2) are employed to generate the rank of each gene. Specifically, N(1,σ2) is for the associated genes and N(0,σ2) for the remaining n − d non-associated genes, in either disease 1 or 2. The gene rank is from the value of normal distribution. The variance σ2 in the normal distribution is taken as 0.01, 0.05, 0.1, 0.5 when o = 0; 0.07, 0.09, 0.11, 0.13 when o = 1; 0.1, 0.15, 0.2, 0.25 when o = 5; 0.2, 0.25, 0.3, 0.35 when o = 10; and 0.3, 0.35, 0.4, 0.45 when o = 20. The variance σ2 plays an important role in discerning the associated genes. The number of replications R is set to 1,000 in the computation of powers/type I error rates, and the nominal significance level is set to 0.05. We set S = 1,000 in the simulation study and S = 10 millions in real data analysis of The Cancer Genome Atlas (TCGA). As an illustration, the empirical distribution of SimSIP for total genes number n = 6,000, 11,000, 25,000 are given in Figs. S1–S3 and the empirical distribution of MAG for n = 6,000, 11,000, 25,000 are given in Figs. S4–S6.

Type I error

We firstly show the simulation results under the null model in evaluating the type I error rates of SimSIP and five existing methods under various scenarios. For the null setting, the empirical p values of existing rank-based methods (WeiSumE*, OrderedList, FES0.001, FES0.01) are computed by permutating ranked gene lists from the value of the normal distribution, and the null distribution of EucD is obtained by permutating the value of the normal distribution generated. All empirical sizes shown in Table S1 are around the significance 0.05 and are well controlled.

Power comparison

In the second set of simulations, the performance of SimSIP is compared with five other existing measures in terms of power. The results show that the powers of six measures are all monotonic decreasing for higher noise (variance σ2) and monotonic increasing for a bigger ratio of o to d. The findings are shown in Fig. 1. Firstly, for the same total gene number n, the smaller the number of associated genes d becomes, the more obvious the advantage of the SimSIP is. For example, in the scenarios of d = 1 (in Figs. 1A, 1E and 1I), the power of SimSIP has almost a 10% increase compared with that of WeiSumE* (Chen, Zhou & Sun, 2015) and OrderedList (Yang et al., 2006), is more than two times the powers of FES0.01 (Serra, Romualdi & Fogolari, 2016), and almost more than 8 times the powers of FES0.001 (Serra, Romualdi & Fogolari, 2016) and EucD. The scenario of d = 5 is analogy to the scenario of d = 1. However, in the scenarios of increasing d up to 10 and 20, the advantage of SimSIP is no longer obvious. Secondly, for the same d, the larger the number of genes (n) becomes, the more obvious the advantages of SimSIP is. For example, in the scenarios of n = 6,000 and n = 11,000, the similarity measure WeiSumE* is the most powerful when the number of associated genes d = 20; however, SimSIP becomes superior when compared with the other five measures when n increases to 25,000. Thirdly, when the number of associated genes d is relatively large, SimSIP gradually manifests certain advantages with the increase in noise (variance σ2), which can be observed in Figs. 1C, 1D, 1G, 1H, 1K and 1L (in the scenarios of d = 10 and d = 20). When increasing the number of overlapping associated genes o up to 5, SimSIP always maintains the superiority compared with the other five methods, and its power has obvious advantages for the larger ratio of o to d with a fixed n and for the larger n with a fixed d (see Fig. 2). When the number of overlapping associated genes o increases from 5 to 10 or 20, the advantages of SimSIP are much more significant (see Figs. S7 and S8).

Figure 1 Powers of EucD, FES0.001, FES0.01, OrderedList, SimSIP and WeiSumE* when o = 1 with 12 scenarios.

(A) n = 6000, d = 1; (B) n = 6000, d = 5; (C) n = 6000, d = 10; (D) n = 6000, d = 20; (E) n = 11000, d = 1; (F) n = 11000, d = 5; (G) n = 11000, d = 10; (H) n = 11000, d = 20; (I) n = 25000, d = 1; (J) n = 25000, d = 5; (K) n = 25000, d = 10; (L) n = 25000, d = 20. The arrangement of variance σ2 on x axis is a series (0.07, 0.09, 0.11, 0.13) on which the power of six measures can be ranged from 0.1 to 0.95.

Figure 2 Powers of EucD, FES0.001, FES0.01, OrderedList, SimSIP and WeiSumE* when o = 5 with nine scenarios.

(A) n = 6000, d = 5; (B) n = 6000, d = 10; (C) n = 6000, d = 20; (D) n = 11000, d = 5; (E) n = 11000, d = 10; (F) n = 11000, d = 20; (G) n = 25000, d = 5; (H) n = 25000, d = 10; (I) n = 25000, d = 20. The arrangement of variance σ2 on x axis is a series (0.1, 0.15, 0.2, 0.25) on which the power of six measures can be ranged from 0.1 to 0.95.

As shown in Figs. 1 and 2 and Figs. S7 and S8, the similarity measure WeiSumE* performs better than other measures when there are less but significant overlap between two diseases with lower noise (variance σ2), and our method SimSIP performs better in all the other scenarios. Especially for n = 25,000, which approximates the number of genes in the full human genome, the performance of SimSIP is always the best. Thus, compared with other measures, SimSIP is more appropriate for detecting genetic similarity between longer gene lists and works well when more overlapping genes occur among diseases. Most importantly, our method SimSIP is more robust for higher noise than the other five methods. Therefore, SimSIP is more suitable for the study of human diseases than the existing methods, especially for the study of cancers in which there are more genetic overlaps.

Furthermore, the similarity measure OrderedList has a higher power with fewer overlaps (for o = 1) and FES0.01 has a higher power with more overlaps (for o > 1). Their performances are superior to those of similarity measure FES0.001 and Euclidean distance EucD, which always have low power in all scenarios. We also find that the rank-based measures always perform better than Euclidean distance EucD, which uses the value of the normal distribution rather than gene ranks. These results demonstrate that the rank of a gene can provide additional information on its contribution to each disease upon converting real data to ranks. In general, compared with existing similarity measures, SimSIP performs better in almost every simulation, which indicates that it is feasible to construct similarity measure by replacing overlaps between top ranked gene sets with gene ranks.

TCGA data analysis

The Cancer Genome Atlas is a combined effort by multiple research institutes, in which tumor and normal samples from more than 11,000 patients are publicly available, comprising 37 types of (epi)genetic and clinical data for 33 types of cancer. We download gene expression datasets (whose gene expression profiles were determined experimentally using the Illumina HiSeq 2000 RNA Sequencing platform) of 33 types of cancer using the UCSC Xena functional genome browser (https://xenabrowser.net/datapages/). Expression data of 20,530 genes are available for each cancer, and 18 types of cancer are selected (see details in Table 1) based on the criteria that the sample size of the control group is not smaller than 5. We rank all genes by their p values derived from the R package LIMMA for differential expression analysis and apply our proposed method to gauge the genetic similarity among the 18 types of cancer in TCGA. In this section, we choose WeiSumE* (Chen, Zhou & Sun, 2015), which performs only second to SimSIP under all simulation scenarios, to compare with SimSIP.

Table 1 Sample sizes of control group and case group per cancer type in TCGA

Abbreviation	Cancer typea	n0b	n1	
BLCA	Bladder urothelial carcinoma	19	407	
BRCA	Breast invasive carcinoma	114	1,097	
CHOL	Cholangiocarcinoma	9	36	
COAD	Colon adenocarcinoma	41	286	
ESCA	Esophageal carcinoma	11	184	
GBM	Glioblastoma multiforme	5	154	
HNSC	Head and neck squamous cell carcinoma	44	520	
KICH	Kidney chromophobe	25	66	
KIRC	Kidney renal clear cell carcinoma	72	533	
KIRP	Kidney renal papillary cell carcinoma	32	290	
LIHC	Liver hepatocellular carcinoma	50	371	
LUAD	Lung adenocarcinoma	59	515	
LUSC	Lung squamous cell carcinoma	51	502	
PRAD	Prostate adenocarcinoma	52	497	
READ	Rectum adenocarcinoma	10	94	
STAD	Stomach adenocarcinoma	35	415	
THCA	Thyroid carcinoma	59	505	
UCEC	Uterine corpus endometrial carcinoma	24	176	
Notes:

a From 33 types of cancer in TCGA, 18 types of cancer with control group sample size being five or more are selected.

b n0 is sample size of control group and n1 is sample size of case group for a given cancer.

For the (18 × 17)/2 = 153 cancer pairs among 18 cancers, we compute their similarity scores by using SimSIP and WeiSumE* respectively and standardize them by normalization method T−TminTmax−Tmin. The normalized similarity scores about SimSIP are in [0.11, 0.77] with mean 0.24 and variance 0.015, and that about WeiSumE* are in [0.064, 0.53] with mean 0.13 and variance 0.005. Obviously, the normalized similarity scores about SimSIP are generally bigger and more spread than those about WeiSumE*. So SimSIP is more discriminative than WeiSumE* in quantifying the relationships among diseases.

Significant cancer pairs in TCGA

To illustrate the application of SimSIP and WeiSumE* in exploring significant relationships among 18 cancers, we compute the empirical p values of SimSIP and WeiSumE* based on the null distribution (shown in Figs. S9 and S10) for the 153 cancer pairs among 18 cancers. Given the nominal significance level of 0.05, the Bonferroni adjustment of empirical p values 0.05/153 is employed to detect significant similar cancer pairs. A total of 91 significant similar cancer pairs are detected by SimSIP, 82 pairs by WeiSumE*, with 81 pairs in their intersection (see Table S2; Fig. S11). For the 81 cancer pairs detected both with SimSIP and WeiSumE* (Chen, Zhou & Sun, 2015), Fig. 3 displays the empirical p values of the two methods for the same cancer pair. Clearly, the empirical p values of SimSIP are generally smaller than that of WeiSumE* (except for cancer pair KIRP and LUAD with index 63 in Fig. 3), which is consistent with the results given in the simulation.

Figure 3 The empirical p values of SimSIP (black dot) and WeiSumE* (red square) for 81 cancer pairs found with both SimSIP and WeiSumE*.

As shown in Fig. S11, SimSIP finds more possibly significant cancer pairs which include almost all the significant cancer pairs found with WeiSumE*. Among the 10 significant cancer pairs (shown in red in Table S2) found with SimSIP, not WeiSumE*, five cancer pairs can be explained in pan-cancer studies: For cancer pair COAD and UCEC, the diversity of high antigen-specific TCR repertoires correlates with the improved prognostic progression-free interval in COAD and UCEC (Thorsson et al., 2018); the expression of TMEM173 in tumor tissues is significantly upregulated and hypomethylated in cancer pair COAD and THCA but significantly downregulated and hypermethylated in cancer pair LUSC and PRAD (An et al., 2019). Furthermore, there are high mutations rates for TBK1 in cancer pair COAD and UCEC, and the expression of TMEM173 is positively associated with the infiltration of immune cells in cancer pair BRCA and THCA (An et al., 2019). A high IRF3 expression yields a poor prediction of prognosis in patients with KIRC and PRAD (An et al., 2019). In addition, cancer pair COAD and UCEC depends on components of the EGFR pathway at similar frequencies (Wormald, Milla & O’Connor, 2013).

Genetic similarity among 18 types of cancer

Depicting the 91 significant similar cancer pairs found with SimSIP in a disease network by using software Cytoscape (https://cytoscape.org/) as in Fig. 4, the following observations could be addressed:

Figure 4 The diseases network of 18 cancer types.

Taking Bonferroni adjustment of empirical p values given nominal significance 0.05 as a threshold, we select 91 significantly similar cancer pairs found with SimSIP to demonstrate the genetic relationships among cancers by software Cytoscape (https://cytoscape.org/). The width, length and color of the edge in the network are determined by the magnitude of SimSIP, the size and color of the vertex in the network is determined by the degree of vertex (the number of cancers which is significantly related to this cancer).

The vertex HNSC has the largest degree, 15, which shows that the HNSC has significant similarity with the other 15 cancers, except for KICH and KIRC. The degrees of vertices such as BLCA and UCEC reach 14; READ, ESCA, LUSC, and HNSC reach 13; and LUAD and LIHC reach 12. There are close intrinsic genetic relationships among the 18 cancers. The seven types of cancers ESCA, LUSC, HNSC, BLCA, BRCA, STAD, and UCEC ( their vertices with deeper color and bigger size) with a higher degree are closer to each other and tend to form a pivotal hub of the disease network of 18 cancer types.

Cancers originating from the same organ or tissue tend to co-cluster, such as cancer pair READ and COAD or KIRP and KIPC with the most similar relationship. In addition, cancers with proximity also tend to group together, such as LIHC and CHOL with a highly significant relationship. These provide evidence that tumors with closer physical distance in human organs have similar sources of endoderm development or exposure to a common cancer-causing environmental factor (Sell & Dunsford, 1989). Further, for the three types of kidney tumors, KICH, KIRP, and KIPC, the similarity between KIRP and KIPC is more significant than KICH and KIPC or KIRP and KIPC, which may be explained by the fact that KIRC and KIRP are cancers of the proximal tubule segments, whereas KICH is a cancer of the distal tubule segments (Chen et al., 2016; Davis et al., 2014; Lee, Chou & Knepper, 2015). Compared to other cancer pairs from the same tissue, the degree of similarity between the two types of lung tumor LUSC and LUAD is not so significant, which may be due to the derivation of cell types of the two types of lung tumors: LUSC originates from squamous epithelial cells in the respiratory tract and alveoli, whereas LUAD originates from a large number of glandular or alveolar cells (Li et al., 2015; Mainardi et al., 2014; Sutherland et al., 2014).

There are significant similarities among gastrointestinal tumors (READ, COAD, STAD, and ESCA), which is consistent with the results of integrative clustering across data types in the miRNA, mRNA, and RPPA platforms (Hoadley et al., 2018). In the disease network (Fig. 4), squamous cell carcinomas (BLCA, ESCA, HNSC, and LUSC) are also co-clustered, and the similarity score of the cancer pair ESCA and LUSC is the top three and has more significant similarity. Hoadley et al. (2018) reported a similar discovery based on a multi-platform dataset (including miRNA, mRNA, RPPA, aneuploidy, and DNA methylation data) in TCGA, and Abrams et al. (2018) also suggested that regardless of the tissue types of squamous cell carcinoma, potential similarities were detected among the transcription factor expression profiles of BLCA, ESCA, HNSC, and LUSC. In addition, the gynecologic tumors UCEC and BRCA are also close to each other in the network, which is consistent with the results of previous studies (Hoadley et al., 2018).

Finally, similar to previous studies (Abrams et al., 2018; Hoadley et al., 2018), three types of cancer, PRAD, THCA, and GBM, are relatively independent in the network, the relationships among them and other cancers are relatively weak. In addition, instead of squamous cancers being clustered together, adenocarcinomas (PRAD, COAD, LUAD, STAD, and READ) appear to be scattered around the edge of the network.

From the above observations of the disease network (Fig. 4) obtained with SimSIP, we find them to be analogous to those of previous pan-cancer studies (Abrams et al., 2018; Hoadley et al., 2018): histologically or anatomically related cancers tend to cluster together, which provide basic support for analyses of pan-gastrointestinal, pan-squamous, pan-kidney, and pan-gynecological cancers.

Associated genes with colorectal cancer

For the most significantly similar cancer pair COAD and READ found with SimSIP, we try to explore the common underlying genetic mechanisms by sorting MAG values of genes in descending order. The significance of the difference of expression level of the top 5 genes in COAD and READ are shown in Table S3, which are obtained from the TCGA data mining website UALCAN (http://ualcan.path.uab.edu/). Table S3 provides evidence that the top 5 genes are associated with COAD and READ and they are regulated in the same pattern (either up-regulated or down-regulated). It is worth mentioning that, for the top 500 genes associated with COAD and READ and the top 500 genes associated with KICH and STAD, which are the most similar and the least similar cancer pairs found with SimSIP respectively, the proportion of genes regulated in the same pattern is 1 and 0.648 respectively. Furthermore, we compute the correlation coefficient between the logarithmic transformation of SimSIP and the proportion of genes regulated in the same pattern in the top N (N = 500, 1,000, 1,500, 2,000) genes associated with each of 153 cancer pairs (shown in Fig. S12). Obviously, there is a significant relationship between the degree of overlap between diseases and the proportion of genes regulated in the same pattern. These findings suggest that the associated genes of highly overlapping cancers may be regulated in the same pattern. Through gene annotation by using Metascape (http://metascape.org/) and literature review, among the top 5 genes, one gene (CDH3) is associated with multiple cancers (shown in Fig. S13), and two genes (ETV4, KRT24) are associated with colorectal cancer; the remaining genes can be considered as candidate susceptibility genes of colorectal cancer (Table S4). CDH3, the top 1 gene detected by MAG, is located in a region on the long arm of chromosome 16. Paredes et al. (2012) advised that genetic or epigenetic changes in this gene or changes in its protein expression often lead to tissue disorders, cellular dedifferentiation, and enhanced invasiveness of tumor cells. This gene is associated with intestinal infections (Van Marck et al., 2011) and colon cancer (Van Marck et al., 2011; Sun et al., 2011). In addition, its over-expression is also associated with tumor progression and low survival in non-small-cell lung cancer (Imai et al., 2018) and in the loss of heterozygous events for breast and prostate cancer (Royo et al., 2016; Sousa et al., 2014; Vieira et al., 2017). Moreover, CDH3 is significantly associated with liver cancer, gastric cancer, bladder cancer, and cervical adenocarcinoma (Bauer et al., 2014; Paredes et al., 2012; Sun et al., 2011; Van Marck et al., 2011). Searching for CDH3 in the CancerMine database (https://www.mycancergenome.org/) reveals that 12 cancers are associated with the over-expression of CDH3. This gene is therefore very important in the genetic mechanism of cancer. The top 3 gene ETV4 is strongly linked to metastasis of colorectal cancer (Dumortier et al., 2018) and enriched in pathway transcriptional misregulation in cancer (Table S4). The top 5 gene KRT24 is over-expressed in patients with colorectal cancer and is a susceptibility gene for early onset of colorectal cancer (Hong et al., 2007).

Associated signaling pathway with colorectal cancer

Calculating empirical p values of MAG for each gene by its null distribution (described in Fig. S14), 1,838 genes significantly associated with colorectal cancer are obtained. By DAVID (https://david.ncifcrf.gov/tools.jsp), these 1,838 genes associated with colorectal cancer are clustered in diverse functional categories.

As shown in Table 2, of 21 functional categories, 3 are associated with cancer (cAMP signaling pathway (Akgul, 2009; Myklebust et al., 1999), fatty acid degradation (Currie et al., 2013), and nitrogen metabolism (Sanchez-Vega et al., 2018)) and 11 appear in colorectal cancer studies, including calcium signaling pathway (Dallol et al., 2003), circadian entrainment (Lévi et al., 2010), ribosome biogenesis in eukaryotes (Lafontaine, 2015), cGMP-PKG signaling pathway (Li et al., 2013), dopaminergic synapse (Xu et al., 2010), retrograde endocannabinoid signaling (Proto et al., 2012), cholinergic synapse (Frucht et al., 1999), insulin secretion (Giovannucci, 2001), gastric acid secretion (Morton, Prendergast & Barrett, 2011), morphine addiction (Jin et al., 2012), and nicotine addiction (Shureiqi et al., 2003; Yang & Frucht, 2001). The remaining signaling pathways can be considered as candidates for studies on biological processes associated with colorectal cancer. In detail, the circadian entrainment pathway closely interacts with the cell division cycle and pharmacological pathways in the treatment of metastatic colorectal cancer and accelerates or slows down cancer growth through modifications of host and tumor circadian clocks, which drives 24 h changes in drug metabolism, cellular proliferation and apoptosis, cell cycle events, DNA repair, and angiogenesis (Lévi et al., 2007, 2010). For the insulin secretion pathway, because insulin and insulin-like growth factor axes are major determinants of cell proliferation and apoptosis, an increase in their circulating concentrations is associated with a high risk of colonic neoplasia. However, the dietary pattern with high saturated fatty acid intake can stimulate insulin resistance or secretion (Giovannucci, 2001), and cellular proliferation requires fatty acids to synthesize cell membranes and signaling molecules (Currie et al., 2013). In addition, the growth of colon tumor cells is selectively inhibited by nonsteroidal anti-inflammatory drugs that activate the cGMP/PKG pathway to suppress Wnt/β-catenin signaling (Li et al., 2013). Up to 15% of colorectal cancers are distinguished by DNA microsatellite instability and manifested by the presence of DNA replication errors (Jass et al., 1998).

Table 2 The result of pathway enrichment analysis of colorectal cancer

ID	KEGG pathway	Benjamini	
hsa04713	Circadian entrainment	2.30E−04	
hsa04024	cAMP signaling pathway	2.97E−04	
hsa03008	Ribosome biogenesis in eukaryotes	3.68E−04	
hsa04080	Neuroactive ligand-receptor interaction	7.92E−04	
hsa04020	Calcium signaling pathway	0.003328211	
hsa04723	Retrograde endocannabinoid signaling	0.006342379	
hsa04724	Glutamatergic synapse	0.006620968	
hsa04911	Insulin secretion	0.010800326	
hsa04728	Dopaminergic synapse	0.011092804	
hsa04725	Cholinergic synapse	0.017964832	
hsa04022	cGMP-PKG signaling pathway	0.021342644	
hsa04970	Salivary secretion	0.023476401	
hsa04261	Adrenergic signaling in cardiomyocytes	0.024338123	
hsa00071	Fatty acid degradation	0.029045641	
hsa05032	Morphine addiction	0.036794268	
hsa04978	Mineral absorption	0.038043761	
hsa04971	Gastric acid secretion	0.045576	
hsa04726	Serotonergic synapse	0.045974199	
hsa00910	Nitrogen metabolism	0.047420936	
hsa05033	Nicotine addiction	0.047658725	
hsa05031	Amphetamine addiction	0.048006097	

Discussion

SimSIP is a novel similarity score that measures the genetic relationships among diseases by (1) introducing a suitable transformation of gene ranks that converts gene ranks into the magnitude of association of gene with the disease; and (2) comparing the similarity between gene rank lists in geometric space instead of comparing the overlaps between the top part of ranked gene lists. In this study, three additional tasks are also fulfilled: constructing a disease network of 18 cancers and offering some support for pan-cancer studies; developing a measure MAG to gauge the magnitude of association of an individual gene with two diseases (note that MAG can be generalized to multiple diseases); and finding some important genes and signaling pathways associated with colorectal cancer.

Extensive simulations show that SimSIP has better performance than existing methods in scenarios with larger numbers of overlapping associated genes (o) and larger number of genes (n), whereas in scenarios with smaller o and smaller n, such as o = 1, n = 6,000 or 11,000, WeiSumE* performs better. It is desirable that SimSIP can clearly identify the differences among diseases with more overlapping associated genes. Thus, in real data application, we use SimSIP to measure genetic similarities among cancers based on the differential expression analyses of multiple datasets in TCGA with the R package LIMMA. The results show that SimSIP can find more significant cancer pairs than WeiSumE*, such as cancer pair COAD and UCEC, COAD and THCA, LUSC and PRAD, BRCA and THCA, or KIRC and PRAD demonstrating the usefulness of SimSIP. Furthermore, for the 81 cancer pairs found with both SimSIP and WeiSumE*, the p value of SimSIP is smaller than WeiSumE*, therefore, the SimSIP may be more powerful than WeiSumE* in detecting the significant cancer pairs.

Overall, SimSIP has a simpler form and faster computation (time consumption for six measures is shown in Table S5), is more robust for higher levels of noise, and is more suitable for the study of human diseases than existing methods, especially for the study of cancers in which there are more genetic overlaps. In order to make this conclusion considerably stronger, we extend the range of the simulations to cover bigger number of overlapping associated genes, such as o = 50, 100 and 200. As shown in Figs. S15–S17, SimSIP still performs well or better in the scenarios with bigger ratio of o to d when o = 50, 100 and 200, which is analogous to previous simulations.

Conclusions

This article proposes a similarity score based on the list of gene ranks to measure the genetic relationships among diseases from gene expression data. Our method SimSIP gives a new perspective to detect the genetic relationships among diseases that does not depend on a threshold as fraction enrichment (Serra, Romualdi & Fogolari, 2016) nor on the weighted sum of overlaps between ranked gene lists as OrderedList (Yang et al., 2006).

Similar to other rank-based measures, our method relies on the correctness and scientific quality of the gene ranking. If gene ranking does not reflect the contribution of the gene to each disease, the rank-based measure is not necessarily superior to the commonly used measures. There are many common methods for ranking genes in practice, such as by the magnitude of p values of t-test, of marginal regression analysis and of some methods for detecting differentially expressed genes (such as LIMMA, edgeR and DESeq2 et al.). Although p values derived from the R package LIMMA for differential expression analysis were used in this paper to rank the gene, different researchers can choose different ranking methods based on their specific needs. In summary, in contrast to existing measures that are all based on the number of overlapping genes in top ranked gene lists among diseases, we creatively describe the genetic relationships among diseases from the spatial similarity between the transformation of gene ranks, which provides a new research direction for studies of similarity measures to reveal genetic relationships among diseases.

Supplemental Information

Supplemental Information 1 Figures S1-S17.

Click here for additional data file.

Supplemental Information 2 Tables S1-S5.

Click here for additional data file.

Supplemental Information 3 The function of MAG.

Click here for additional data file.

Supplemental Information 4 The function of SimSIP.

Click here for additional data file.

We thank two anonymous reviewers for their constructive comments and suggestions that improve the presentation of the manuscript greatly. We thank our lab members Liming Li and Yuquan Wang for code guidance, and our colleagues Zhongzhi Wang, Yongjin Zhang, and Wenxi Li for career help.

Additional Information and Declarations

Competing Interests

Author Contributions

Data Availability

The authors declare that they have no competing interests.

Dongmei Luo conceived and designed the experiments, analyzed the data, prepared figures and/or tables, authored or reviewed drafts of the paper, and approved the final draft.

Chengdong Zhang conceived and designed the experiments, prepared figures and/or tables, authored or reviewed drafts of the paper, and approved the final draft.

Liwan Fu analyzed the data, prepared figures and/or tables, and approved the final draft.

Yuening Zhang performed the experiments, prepared figures and/or tables, and approved the final draft.

Yue-Qing Hu conceived and designed the experiments, analyzed the data, authored or reviewed drafts of the paper, and approved the final draft.

The following information was supplied regarding data availability:

Gene expression datasets (whose gene expression profiles were determined experimentally using the Illumina HiSeq 2000 RNA Sequencing platform) for 33 types of cancer are available at the UCSC Xena functional genome browser from the TCGA hub (https://tcga.xenahubs.net).

The names of datasets are TCGA Acute Myeloid Leukemia (LAML), TCGA Adrenocortical Cancer (ACC), TCGA Bile Duct Cancer (CHOL), TCGA Bladder Cancer (BLCA), TCGA Breast Cancer (BRCA), TCGA Cervical Cancer (CESC), TCGA Colon Cancer (COAD), TCGA Endometrioid Cancer (UCEC), TCGA Esophageal Cancer (ESCA), TCGA Glioblastoma (GBM), TCGA Head and Neck Cancer (HNSC), TCGA Kidney Chromophobe (KICH), TCGA Kidney Clear Cell Carcinoma (KIRC), TCGA Kidney Papillary Cell Carcinoma (KIRP), TCGA Large B-cell Lymphoma (DLBC), TCGA Liver Cancer (LIHC), TCGA Lower Grade Glioma (LGG), TCGA Lung Adenocarcinoma (LUAD), TCGA Lung Squamous Cell Carcinoma (LUSC), TCGA Melanoma (SKCM), TCGA Mesothelioma (MESO), TCGA Ocular melanomas (UVM), TCGA Ovarian Cancer (OV), TCGA Pancreatic Cancer (PAAD), TCGA Pheochromocytoma & Paraganglioma (PCPG), TCGA Prostate Cancer (PRAD), TCGA Rectal Cancer (READ), TCGA Sarcoma (SARC), TCGA Stomach Cancer (STAD), TCGA Testicular Cancer (TGCT), TCGA Thymoma (THYM), TCGA Thyroid Cancer (THCA), and TCGA Uterine Carcinosarcoma (UCS).

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
