# Peer review of "A novel similarity score based on gene ranks to reveal genetic relationships among diseases"

_PeerJ, doi:10.7717/peerj.10576_

## Round 0.1 · original submission · Major Revisions

Please, pay attention that both reviewers are very positive and their suggestions have been made to improve your manuscript.

Reviewer 1 ·

Basic reporting

With regard to two diseases, for a list of genes they are associating with these two diseases, when the ranks of these genes are considered in the association with the diseases, the author has proposed to use the geometric mean of the reciprocals of the ranks as the basic unit to measure the similarity between the two diseases by summing up all of the geometric means. The measurement is termed as SimSIP. The new similarity measurement was tested on simulation datasets, and also on real gene expression datasets from TCGA. The method was compared with WeiSumE*, OrderedList, FES0.01 and EucD to demonstrate its superior performance in the identification of similar diseases. The author also reported associated genes and pathways with colorectal cancer.

Experimental design

Comments: The manuscript is well written and organized; the novelty of the ideas is not that strong; the performance evaluation is well designed and discussed.

Validity of the findings

Suggestions:
1. I don’t understand why equation \sum^n_{i=1}(sqrt{1/a_i})^2 is popped up in the second paragraph of page 3. Why did you need this sentence? Please clarify.
2. Ranking methods for genes are less discussed in the method section. In fact, the SimSIP results are very sensitive to the ranking lists of the genes. Which ranking methods should be recommended? Please clarify.

Additional comments

See above.

Reviewer 2 ·

Basic reporting

Generally, the paper makes a good impression mainly connected to the experimental study section. However, the theoretical part raises many questions. First of all, the motivation according to the implication of the new connection measure is very blurred. Computanchilay, a vector composed from the inverse ranges, provides more reliable results, but it is entirely not understanding why. An explanation given in rows 115-118 is very abuse and does not offer any useful information about why such a measure performs better in comparison with the standard inner product.
The paper is written in a deplorable manner. For example, what does it mean:
All empirical sizes shown in Supplementary 158 Table S1 are around the significance 0.05 and are well controlled. I feel that the paper provides exciting results, but it very hard to understand it from the text.

Experimental design

Provided in a good manner

Validity of the findings

I thing that more solid comparison has to be done

Additional comments

Generally, the paper makes a good impression mainly connected to the experimental study section. However, the theoretical part raises many questions. First of all, the motivation according to the implication of the new connection measure is very blurred. Computanchilay, a vector composed from the inverse ranges, provides more reliable results, but it is entirely not understanding why. An explanation given in rows 115-118 is very abuse and does not offer any useful information about why such a measure performs better in comparison with the standard inner product.
The paper is written in a deplorable manner. For example, what does it mean:
All empirical sizes shown in Supplementary 158 Table S1 are around the significance 0.05 and are well controlled. I feel that the paper provides exciting results, but it very hard to understand it from the text.

---

## Round 0.2 · accepted · Accept

Thank you for consideration of the changes proposed by our reviewers.

Reviewer 1 ·

Basic reporting

The basic findings stated in the manuscript have not been changed after the revision. The authors have well addressed my questions and concerns.

Experimental design

The experimental design is sound.

Validity of the findings

The performance evaluation can be trusted.

Additional comments

The revised version of the manuscript has reached a very high standard. I recommend to accept the paper for publication in PeerJ.

Reviewer 2 ·

Basic reporting

The current presentation arrears to be sufficiently appropriate, but intensive proofreading is required.

Experimental design

The experiments are designed well

Validity of the findings

Experiments validity is appropriate

Additional comments

It would help if you improve the English style